# One SPACE to Rule Them All: Jointly Mitigating Factuality and Faithfulness Hallucinations in LLMs

**Pengbo Wang**[†]   **Chaozhuo Li**[†*]   **Chenxu Wang**[‡]   **Liwen Zheng**[†]   **Litian Zhang**[†]   **Xi Zhang**[†]

[†]Beijing University of Posts and Telecommunications

[‡]Shihezi University

{1696280515, lichaozhuo, zhenglw, litianzhang, zhangx}@bupt.edu.cn

axsdsama@gmail.com

## Abstract

LLMs have demonstrated unprecedented capabilities in natural language processing, yet their practical deployment remains hindered by persistent factuality and faithfulness hallucinations. While existing methods address these hallucination types independently, they inadvertently induce performance trade-offs, as interventions targeting one type often exacerbate the other. Through empirical and theoretical analysis of activation space dynamics in LLMs, we reveal that these hallucination categories share overlapping subspaces within neural representations, presenting an opportunity for concurrent mitigation. To harness this insight, we propose SPACE, a unified framework that jointly enhances factuality and faithfulness by editing shared activation subspaces. SPACE establishes a geometric foundation for shared subspace existence through dual-task feature modeling, then identifies and edits these subspaces via a hybrid probe strategy combining spectral clustering and attention head saliency scoring. Experimental results across multiple benchmark datasets demonstrate the superiority of our approach. https://github.com/chronostesis/1-SPACE-2-Rule-Them-All

## 1   Introduction

LLMs have fundamentally transformed the landscape of natural language processing [1, 2]. However, a fundamental challenge remains unaddressed: the persistent manifestation of hallucinations, defined as instances where these models generate text containing factual incongruities or deviations from contextual coherence [3]. This phenomenon becomes particularly consequential when considering application scenarios demanding rigorous epistemological integrity.

Hallucinations can be classified into two primary types: factuality hallucinations and faithfulness hallucinations [3, 4, 5, 6]. Factuality hallucinations refer to contradictions against verifiable facts, whereas faithfulness hallucinations denote deviations from user intent or exhibit internal inconsistencies. As shown in Figure 1(a), claiming "Sydney is the capital of Australia" is a factuality hallucination, while answering "The cheetah runs fastest" to "Who runs faster, the turtle or the rabbit?" is a faithfulness hallucination, since it is correct but unfaithful to the prompt. Both categories significantly compromise the reliability of LLMs, as even isolated instances erode user trust in generated content.

Existing works generally address hallucination categories in isolation, focusing on optimizing either factuality [7, 8, 9] or faithfulness [10, 11, 12] through distinct methodological frameworks. However, our study reveals a critical oversight: interventions targeting one type of hallucination often exacerbate others. As illustrated in Figure 1(b), experiments are conducted on two benchmark datasets:

---

[*]Corresponding author: Chaozhuo Li

39th Conference on Neural Information Processing Systems (NeurIPS 2025).

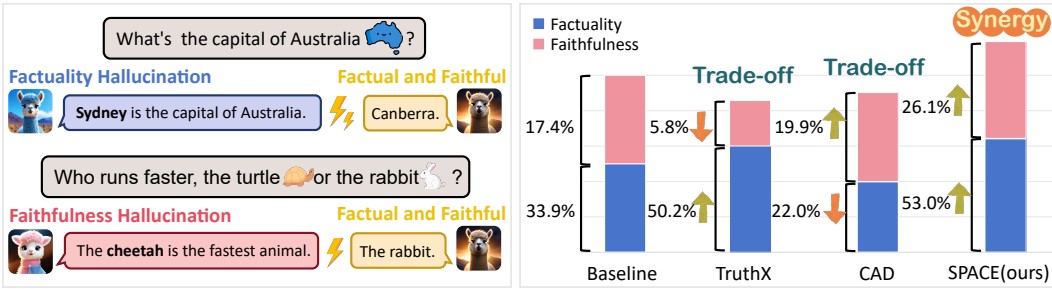

(a) Examples of factuality and faithfulness hallucinations.    (b) Performance on factuality and faithfulness.

Figure 1: The illustrations of two hallucination types and the performance trade-off.

TruthfulQA [13] for factuality evaluation and the PDTB [14] corpus, with DISQ [15] employed as the metric for faithfulness. Two prominent models—TruthX (designed for factual accuracy optimization) and CAD (developed to enhance faithfulness)—are selected as comparative baselines. Compared to the base model (Llama-2-7b), TruthX significantly improves factuality metrics but shows performance degradation when evaluated on the PDTB corpus.Conversely, CAD improves faithfulness on PDTB while sacrificing factuality. These results underscore a clear trade-off between factuality and faithfulness in existing works.

Focusing on the tradeoff between different types of hallucinations, this paper proposes to simultaneously enhance performance for both categories through a unified operation – achieving "killing two birds with one stone". Building on prior research [16, 17], the activation space is adopted as the technical substrate. Formally defined as the comprehensive set of intermediate representations generated by neural networks, the activation space provides crucial insights into the cognitive mechanisms of LLMs [18] and reveals intrinsic correlations with hallucination phenomena [19, 20]. Hallucination mitigation can be systematically interpreted as targeted modifications within particular subspaces of the activation space [21]. Intuitively, the subspaces associated with different hallucination types appear independent due to their distinct definitions [22, 23]. Editing or expanding the subspace related to one type of hallucination may inadvertently distort or compress the subspace of another, thereby creating performance tradeoffs. However, due to the shared parameter architecture of LLMs [24] and interconnected nature of different hallucination types, their subspaces may exist overlapped region. If such shared subspace could be targeted for editing, both hallucination types could be concurrently optimized, resulting in mutual performance enhancement rather than compromise.

While promising, modifying shared activation subspaces to simultaneously improve factuality and faithfulness in LLMs remains challenging. First, the existence of such a shared subspace is often intuitively assumed, yet providing a rigorous theoretical foundation for its validity remains non-trivial. Second, identifying this subspace is inherently difficult due to the high-dimensional activation space of LLMs, which complicates precise localization. Third, even if such a subspace were identifiable, developing an effective editing methodology that modifies the targeted space without compromising model performance introduces additional difficulty.

To address the aforementioned challenges, we propose SPACE (Spatial Processing for Activated Combined Embeddings), a novel framework that steers LLMs toward the shared activation space of factuality and faithfulness. We rigorously formalize the intersection of activation patterns in Section 2.2 through systematic extraction and geometric modeling of dual-task activation features, thereby establishing a mathematical foundation for subspace existence. A hybrid probe strategy is further proposed to combine the spectral clustering with attention head saliency scoring, enabling precise identification of critical subspace-contributing heads. By constructing adjustment vectors derived from the extracted features, SPACE ensures these edits align with both objectives (factuality and faithfulness) without biasing either. These vectors are then applied to the identified attention heads, enabling precise modifications that preserve overall model performance. Experimental results over SOTA baselines demonstrate the superiority of our proposal.

In summary, our contributions are as follows:

- We theoretically reveal a trade-off between mitigating factuality and faithfulness hallucinations in LLMs, which stems from divergent activation subspaces optimized for each task during training.

- To resolve this trade-off, we propose SPACE, a novel model that identifies and edits the shared activation subspace where factuality and faithfulness intersect, resulting in the "two birds with one stone" effect.
- Extensive experiments on standard benchmarks demonstrate SPACE's superiority over SOTA baselines, validating its effectiveness in jointly improving LLM factuality and faithfulness.

## 2 Preliminary Analysis

### 2.1 Empirical Analysis of Hallucination-Related Activation Patterns

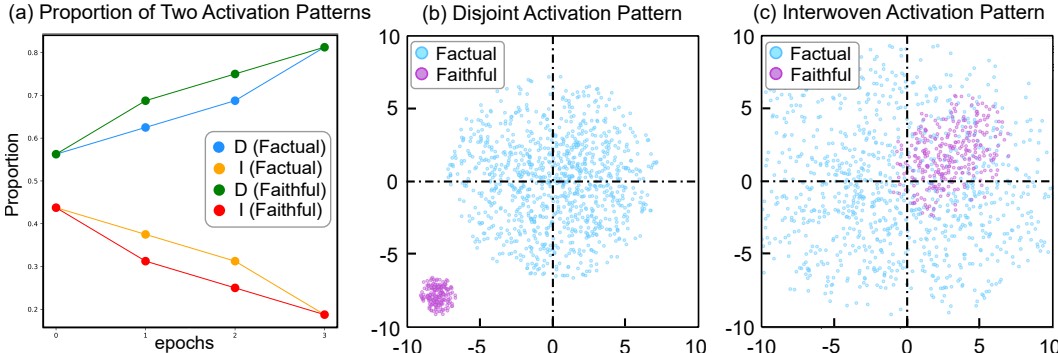

Figure 2: Activation Distributions of Factual and Faithful Tasks. Subfigure (a) shows the proportion of disjoint and interwoven activation pattern after training. Subfigure (b) and (c) show two cases of disjoint activation pattern and interwoven activation pattern respectfully.

Our research commences with a systematic examination of activation dynamics in LLMs concerning their factuality and faithfulness. The study employs Llama3.2-1B as the target model and utilizes TruthfulQA [13] for factual enhancement and PDTB [15] for faithfulness improvement through instruction tuning. In each training epoch, the activations from each layer of the target LLM are collected and visualized via PCA. As illustrated in Figure 2 (b) and (c), we identify two distinct activation patterns: the disjoint activation pattern, which exhibits clear spatial separation between neural activations related to factual and faithful processing, and the interwoven activation pattern, where these two types of activations show significant overlap. During training on the factual dataset, the proportion of disjoint activation patterns (D(Factual)) exhibits a consistent increase, whereas interwoven patterns (I(Factual)) demonstrate a decline. This divergence suggests a progressive separation between the two activation subspaces, indicative of their growing relative distance.

The results imply that activation subspaces corresponding to distinct task types are localized within the model's parameter space. However, introducing additional tasks during training can perturb these subspaces, potentially destabilizing pre-existing representations. A promising direction for mitigating this interference involves identifying and isolating these subspaces, then guiding the LLM to learn features within their intersectional region. Such an approach could enable the model to harmonize and retain the desired attributes of both tasks in its generative outputs, thereby ameliorating the trade-off.

### 2.2 Theoretical Guarantee for the Existence of the Overlapped Subspace

The key to realizing the conjecture formulated in the previous section is to ensure the existence of an intersection between the activation subspaces associated with factuality and faithfulness. We provide a theoretical proof that such an overlapped subspace necessarily exists in LLMs.

Let $\mathcal{V} \subseteq \mathbb{R}^d$ be a complete Hilbert space equipped with the standard inner product $\langle \cdot, \cdot \rangle$. Define the factuality subspace $\mathcal{F}_{\text{fact}} \subseteq \mathcal{V}$ and the faithfulness subspace $\mathcal{F}_{\text{faith}} \subseteq \mathcal{V}$ as two nonempty, closed, and convex subsets of $\mathcal{V}$. Additionally, suppose that these sets satisfy the following convex interpolation property: for any $\mathbf{v} \in \mathcal{F}_{\text{fact}}$ and $\mathbf{u} \in \mathcal{F}_{\text{faith}}$, and for any $\lambda \in [0, 1]$, the convex combination

$$\mathbf{w} = \lambda \mathbf{v} + (1 - \lambda)\mathbf{u}$$

also belongs to $\mathcal{F}_{\text{fact}} \cap \mathcal{F}_{\text{faith}}$. This assumption reflects the preservation of both factuality and faithfulness properties under linear interpolation.

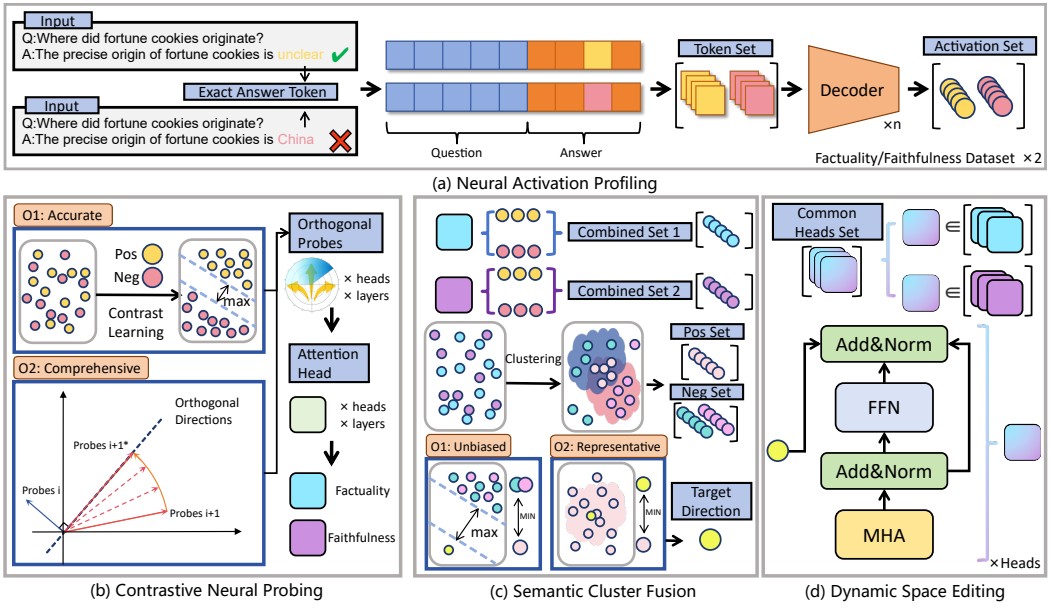

Figure 3: Framework of the proposed SPACE model.

To establish the existence of an intersection between $\mathcal{F}_{\text{fact}}$ and $\mathcal{F}_{\text{faith}}$, suppose, for contradiction, that $\mathcal{F}_{\text{fact}} \cap \mathcal{F}_{\text{faith}} = \emptyset$. By the Hahn–Banach Separation Theorem [25], there exists a nonzero vector $\mathbf{a} \in \mathcal{V}$ and a constant $\gamma \in \mathbb{R}$ such that

$$\langle \mathbf{a}, \mathbf{v} \rangle \leq \gamma < \langle \mathbf{a}, \mathbf{u} \rangle, \quad \forall \mathbf{v} \in \mathcal{F}_{\text{fact}}, \quad \forall \mathbf{u} \in \mathcal{F}_{\text{faith}}.$$

This implies that every point in $\mathcal{F}_{\text{fact}}$ lies entirely in one half-space defined by $\mathbf{a}$, while every point in $\mathcal{F}_{\text{faith}}$ lies strictly in the other half-space.

Now, consider an arbitrary $\mathbf{v} \in \mathcal{F}_{\text{fact}}$ and $\mathbf{u} \in \mathcal{F}_{\text{faith}}$. By the convex interpolation assumption, for any $\lambda \in (0, 1)$, the point

$$\mathbf{w} = \lambda \mathbf{v} + (1 - \lambda)\mathbf{u}$$

must belong to $\mathcal{F}_{\text{fact}} \cap \mathcal{F}_{\text{faith}}$. However, applying the separating hyperplane inequality, we obtain

$$\langle \mathbf{a}, \mathbf{w} \rangle = \lambda \langle \mathbf{a}, \mathbf{v} \rangle + (1 - \lambda)\langle \mathbf{a}, \mathbf{u} \rangle.$$

Since $\langle \mathbf{a}, \mathbf{v} \rangle \leq \gamma$ and $\langle \mathbf{a}, \mathbf{u} \rangle > \gamma$, the convex combination satisfies

$$\lambda \langle \mathbf{a}, \mathbf{v} \rangle + (1 - \lambda)\langle \mathbf{a}, \mathbf{u} \rangle \leq \lambda\gamma + (1 - \lambda)\langle \mathbf{a}, \mathbf{u} \rangle.$$

For sufficiently small $\lambda$, this implies

$$\langle \mathbf{a}, \mathbf{w} \rangle < \langle \mathbf{a}, \mathbf{u} \rangle,$$

which contradicts the requirement that $\mathbf{w} \in \mathcal{F}_{\text{faith}}$, as it should satisfy $\langle \mathbf{a}, \mathbf{w} \rangle \geq \gamma$. This contradiction invalidates the assumption that $\mathcal{F}_{\text{fact}}$ and $\mathcal{F}_{\text{faith}}$ are disjoint, proving that

$$\mathcal{F}_{\text{fact}} \cap \mathcal{F}_{\text{faith}} \neq \emptyset.$$

Thus, the existence of an optimal subspace where both factuality and faithfulness are simultaneously satisfied is guaranteed.

## 3   Methodology

As shown in Figure 3, the SPACE comprises four major stages. First, neural activation profiling derives activation values from each attention head using a dataset containing questions paired with both correct and incorrect answers. Second, contrastive neural probing employs probes trained with a contrastive loss function to identify neurons associated with factuality and faithfulness, while applying orthogonality constraints to enhance probe independence. Third, semantic cluster fusion utilizes the HDBSCAN clustering algorithm [26] to integrate factuality and faithfulness activation data, precisely constructing target direction vectors. Finally, the Dynamic Space Editing adaptively adjusts neuron parameters based on preceding stage outputs.

## 3.1 Neural Activation Profiling

To formalize the neural activation profiling for a given dataset comprising questions, non-hallucinated answers, and hallucinated answers, the following procedure is implemented. For each question $q$, it is concatenated with its corresponding non-hallucinated answer $a^+$ and hallucinated answer $a^-$, respectively. An instruction $I$, designed to prompt the extraction of precise answer tokens, is prepended to each concatenated pair. This yields the input dataset:

$$D_{\text{in}} = \{(I, q, a^+), (I, q, a^-)\}$$

The inclusion of this instruction is motivated by its empirically validated capacity to enhance the model's focus on task-relevant semantic representations, as demonstrated in prior work on instruction-driven feature extraction [27]. Subsequently, the activation data generated by the LLM during inference over $D_{\text{in}}$ are collected for analysis. The updating rule of the $l^{\text{th}}$ layer in a decoder-only LLM is as follows:

$$x^{l+\frac{1}{2}} = \text{Norm}_1\left(x^l + \text{MHA}(x^l)\right)$$

$$x^{l+1} = \text{Norm}_2\left(x^{l+\frac{1}{2}} + \text{FFN}(x^{l+\frac{1}{2}})\right)$$

where $x^l$ represents the $l$-th layer, $\text{Norm}_1$ and $\text{Norm}_2$ are normalization functions, MHA stands for multi-head attention, and FFN denotes the feed-forward network. Each input prompt in $D_{in}$ is first embedded into $x^0$, and then updated in subsequent layers. Next, $x^{l+\frac{1}{2}}$ in each layer is collected and partitioned by attention heads. We construct the activation dataset $D_{out} = \{[(A^+)_l^h, (A^-)_l^h]\}$, which consists of the activations of each head in each layer: $x_+^{l,h}$ derived from $(I, q, a^+)$, and $x_-^{l,h}$ derived from $(I, q, a^-)$. This procedure is applied to both the factuality and faithfulness datasets.

## 3.2 Contrastive Neural Probing

Following the collection of activation patterns from both the factuality and faithfulness datasets, we employ these activations to train diagnostic probes for every attention head across all layers of the LLM, which nables the identification of neurons correlated with the target capability. To enhance the model's discriminative capacity between correct and incorrect statements, a contrastive loss function is designed to maximize the separation in the latent space between the representations of positive and negative samples, thereby sharpening the model's ability to discern veracity: For each pair $(x_i, x_j)$, the contrastive loss is defined as:

$$\mathcal{L}_{\text{ctr}} = -\frac{1}{N} \sum_{i=1}^{N} [y_i \log(p_\theta(x_i, x_j)) + (1 - y_i) \log(1 - p_\theta(x_i, x_j))]$$

where $y_i$ denotes the label, and $p_\theta(x_i, x_j)$ is the probability that the pair $(x_i, x_j)$ is true, calculated using the sigmoid function:

$$p_\theta(x_i, x_j) = \sigma(\langle f(x_i), f(x_j)\rangle).$$

Here, $f(x)$ represents the feature representation obtained from the probe activation.

To more effectively capture the desired subspace that represents truth [28], we utilize multiple probes with directions spanning higher-dimensional space. Each probe $\theta_i \in \Theta = \{\theta_1, \theta_2, \ldots, \theta_n\}$ is designed to focus on a distinct aspect of the model representation. The probes are orthogonal, that is, $\theta_i \perp \theta_j$ for $i \neq j$, ensuring that each probe captures independent complementary features of the model's internal representation of factuality or faithfulness,which reduces redundancy and promotes diversity in learned features. The orthogonality loss is formalized as:

$$\mathcal{L}_{\text{orth}} = \sum_{i=1}^{k} \sum_{j=1}^{i-1} \sigma(\langle \theta_i, \theta_j\rangle)$$

where $\sigma(\langle \theta_i, \theta_j\rangle)$ is the sigmoid function applied to the dot product of the probe vectors, encouraging orthogonality or independence between probes. This allows for flexibility in adjusting the strength of the orthogonality constraint, depending on the similarity between probe vectors.

To adaptively control the strength of the orthogonality loss, we introduce a dynamic adjustment $\lambda_t$, the weight applied to the orthogonality loss. This adjustment is based on the gradient of the

orthogonality loss, ensuring that $\lambda_t$ increases when the gradient is large and decreases when the gradient is small. This dynamic adjustment ensures that the orthogonality constraint is applied more rigorously when needed, without dominating the training process. The adjustment rule is defined as:

$$\lambda_t = \lambda_0 \cdot \left(1 + p \cdot \frac{\|\nabla_{\theta_i} L_{\text{orth}}\|^2}{\|\nabla \theta_i L_{\text{orth}}\|_{\max}}\right)$$

where $p$ is a scaling factor, and $\|\nabla_{\theta_i} L_{\text{orth}}\|_{\max}$ is the maximum gradient across all probes. This dynamic modulation ensures that the orthogonality constraint is appropriately applied.

Finally, the total loss function combining both the contrastive and orthogonality losses is:

$$\mathcal{L}_{\text{total}} = \mathcal{L}_{\text{ctr}} + \lambda_t \mathcal{L}_{\text{orth}}$$

This formulation ensures that the model is optimized to effectively distinguish between factual statements and hallucinated ones, enabling accurate identification of relevant neurons. After training the probes, we employ them to extract the top-k heads associated with factuality and faithfulness, respectively, then determine their intersection to pinpoint co-activated neurons.

### 3.3 Semantic Cluster Fusion

Semantic cluster fusion aims to merging representations related to factuality and faithfulness to train the target direction vectors $\theta_l^h$. We begin by applying the HDBSCAN algorithm to cluster embeddings separately for factuality and faithfulness. A point $x$ is considered a core point if there are sufficient points within its $\epsilon$-neighborhood $N_\epsilon(x)$, defined as:

$$N_\epsilon(x) = \{y \mid d(x, y) \leq \epsilon\}.$$

Clusters are formed by expanding from core points and including all points that are density-reachable. The clustering of factuality embeddings $X_{\text{factual}}$ and faithfulness embeddings $X_{\text{faithful}}$ results in:

$$X_{\text{factual}} \rightarrow \{C_{\text{factual}}^1, C_{\text{factual}}^2, \ldots C_{\text{factual}}^{N_{\text{factual}}}\}$$

$$X_{\text{faithful}} \rightarrow \{C_{\text{faithful}}^1, C_{\text{faithful}}^2, \ldots, C_{\text{faithful}}^{N_{\text{faithful}}}\}$$

Here, $C_{\text{factual}}^i$ and $C_{\text{faithful}}^j$ denote clusters of factuality and faithfulness embeddings, respectively. From these clusters, positive and negative samples are defined as follows:

$$P = \{x \mid x \in C_{\text{factual}}^i \cap C_{\text{faithful}}^j, \forall i, j\}$$

$$N = \{x \mid x \notin C_{\text{factual}}^i \cap C_{\text{faithful}}^j, \forall i, j\}$$

Positive samples $P$ are embeddings in the overlap of clusters, exhibiting high similarity in both factuality and faithfulness, while negative samples $N$ represent embeddings outside this overlap. The target direction $\mathbf{d}$ denotes the difference between the centroids of positive and negative samples:

$$\mathbf{d} = \frac{1}{|P|} \sum_{x^+ \in P} f(x^+) - \frac{1}{|N|} \sum_{x^- \in N} f(x^-)$$

A contrastive learning framework with hard negative sampling is proposed to train this module as:

$$\mathcal{L}_{\text{hard-ctr}} = \frac{1}{N} \sum_{i=1}^{N} \max\left(0, \|f(x_i) - \mathbf{d}\|_2^2 - \max_j \|f(x_i) - f(x_j^-)\|_2^2 + \text{margin}\right)$$

where $f(x_i)$ is the embedding of the anchor $x_i$, and $f(x_j^-)$ represents the $j$-th negative sample. The term $\max_j \|f(x_i) - f(x_j^-)\|_2^2$ identifies the hardest negative sample closest to $x_i$, ensuring the model focuses on challenging cases. The margin term enforces a minimum separation between positive and negative samples. In this framework, anchors $x_i \in P$, while $x_j^- \in N$. Hard negative sampling emphasizes distinctions between $P$ and $N$, refining the embedding space to better capture the nuances of factuality and faithfulness.

Table 1: Cross-model comparison of editing methods on TruthfulQA and PDTB with DISQ evaluation.

| Method | Model | PDTB | | | | TruthfulQA | | | | |
|---|---|---|---|---|---|---|---|---|---|---|
| | | Overall | Targeted | Counterfactual | Consistency | True*Info | MC1 | MC2 | True | Info |
| Baseline | | 15.1 | 21.2 | 84.0 | 85.1 | 59.5 | 37.9 | 55.7 | 70.1 | 84.8 |
| FT—PDTB | deepseek-llm-7b | 20.1 | 27.8 | 86.2 | 83.7 | 46.8 | 33.1 | 46.2 | 65.1 | 71.8 |
| FT—TruthfulQA | | 3.5 | 5.0 | 82.4 | 83.9 | 64.3 | 49.4 | 67.9 | 73.3 | 87.8 |
| CAD | | 15.6 | 23.4 | 80.9 | 82.4 | 49.3 | 33.3 | 54.9 | 70.0 | 70.5 |
| DoLa | | 10.1 | 14.6 | 83.1 | 83.4 | 60.1 | 29.9 | 7.5 | 70.3 | 85.5 |
| ITI | | 15.7 | 22.2 | 83.5 | 84.9 | 62.9 | 38.6 | 57.9 | 71.3 | 88.2 |
| SPACE | | **23.0** | **29.5** | **89.1** | **87.4** | **71.9** | **51.0** | **68.1** | **79.4** | **90.5** |
| Baseline | | 26.5 | 38.0 | 81.5 | 85.7 | 54.8 | 41.1 | 61.3 | 77.7 | 70.5 |
| FT—PDTB | Qwen2-7B-Instruct | 27.8 | 39.3 | 82.1 | 86.3 | 46.1 | 37.4 | 53.1 | 72.8 | 63.4 |
| FT—TruthfulQA | | 15.2 | 24.2 | 80.2 | 78.4 | 64.6 | 53.1 | 73.6 | 83.1 | 77.7 |
| CAD | | 33.2 | 47.6 | 84.3 | 82.7 | 50.8 | 36.1 | 53.8 | 74.0 | 68.6 |
| DoLa | | 15.0 | 21.9 | 81.3 | 84.2 | 61.3 | 34.2 | 29.6 | 83.2 | 73.6 |
| ITI | | 17.3 | 29.5 | 70.4 | 83.5 | 56.2 | 43.6 | 63.8 | 78.2 | 71.8 |
| SPACE | | **44.6** | **53.9** | **90.6** | **91.3** | **75.8** | **57.2** | **78.0** | **89.7** | **84.5** |
| Baseline | | 17.4 | 79.3 | 27.1 | 81.2 | 57.6 | 33.9 | 51.3 | 66.9 | 86.1 |
| FT—PDTB | LLaMA2-7B-Chat | 23.7 | 81.5 | 34.9 | 83.2 | 46.1 | 30.7 | 45.1 | 62.8 | 73.3 |
| FT—TruthfulQA | | 12.7 | 69.2 | 22.7 | 80.6 | 62.5 | 49.7 | 63.2 | 69.0 | 90.6 |
| CAD | | 20.0 | 70.5 | 35.4 | 80.1 | 54.2 | 22.0 | 50.8 | 69.0 | 78.5 |
| DoLa | | 15.9 | 78.0 | 25.1 | 81.2 | 59.2 | 33.3 | 60.9 | 67.6 | 87.5 |
| ITI | | 14.8 | 64.5 | 28.7 | 79.8 | 59.9 | 33.9 | 52.0 | 68.9 | 87.0 |
| TruthX | | 5.8 | 75.6 | 9.6 | 80.2 | 63.6 | 50.2 | 70.5 | 70.8 | 89.7 |
| SPACE | | **26.1** | **82.3** | **35.5** | **89.2** | **67.1** | **53.0** | **72.6** | **71.2** | **94.2** |

## 3.4 Dynamic Space Editing

In this section, SPACE utilizes the previously learned direction vectors $\theta_l^h$ to adjust the parameters of the selected neurons within each layer of the Transformer model. This adjustment refines the model's focus on the most relevant features identified during the selection process, ensuring a more targeted and effective representation while maintaining training stability. The key idea behind this dynamic adjustment is to modify the output of each layer based on the contribution of individual attention heads and their corresponding learned directions. The update rule for the neuron parameters is:

$$x^{l+1} = \text{Norm}_2\left(x^{l+\frac{1}{2}} + \text{FFN}(x^{l+\frac{1}{2}}) + \alpha \sum_{h=1}^{H} s_l^h \theta_l^h\right).$$

The term $\alpha \sum_{h=1}^{H} s_l^h \theta_l^h$ is the dynamic adjustment factor, where $s_l^h$ is the standard deviation of $(A^+)_l^h$ and $(A^-)_l^h h$ along $\theta_l^h$, and $\theta_l^h$ is the direction vector associated with that head. The hyperparameter $\alpha$ controls the magnitude of this adjustment.

## 4 Experiments

### 4.1 Experimental Settings

**Datasets and Metrics.** Our framework is validated on two benchmarks: PDTB [29] evaluated via DISQ Score [15], measuring discourse understanding through Targeted (event accuracy), Counterfactual (robustness), and Consistency (logical coherence) components, with an Overall product score. TruthfulQA [13] assesses truthfulness using MC1 (single-answer accuracy), MC2 (multi-answer probability), Truthfulness (avoiding falsehoods), and Informativeness (response quality), with true*informative as the composite metric.

**Implementation Details.** The SPACE model is trained on a 20% sampled dataset, with the remaining data reserved for evaluation. Implemented in PyTorch, the model leverages GPU-accelerated mixed-precision training with BF16 to optimize both computational efficiency and memory utilization. For evaluating "Truth" and "Info" metrics, we employ two LLaMA-2-7B models fine-tuned on

Table 2: Generalization capacity of our proposal.

| Methods | PDTB | TruthfulQA | | |
|---|---|---|---|---|
| | Overall (%) | True*Info (%) | MC1 (%) | MC2 (%) |
| Alpaca2_7B | 16.3 | 47.2 | 33.4 | 50.3 |
| +SPACE | 18.4 | 54.0 | 35.0 | 52.8 |
| Llama-3-Chinese-8B-Instruct-v3 | 11.1 | 62.3 | 38.8 | 56.7 |
| +SPACE | 13.2 | 69.6 | 39.3 | 58.1 |
| MiniCPM-1B-SFT | 16.2 | 35.4 | 24.4 | 39.4 |
| +SPACE | 18.6 | 38.4 | 25.5 | 44.3 |

TruthfulQA benchmark's dedicated datasets. The reported performance metrics represent the best results from published literature, supplemented by our local experimental findings.

## 4.2 Main Results

As demonstrated in Table 1, SPACE exhibits consistent superiority over baseline methods across all evaluation metrics. Notably, the model achieves concurrent improvements in both factuality and faithfulness across all evaluated datasets, without observable trade-offs between these two dimensions—a capability not observed in competing methods. While alternative approaches demonstrate localized enhancements in specific sub-metrics (suggesting partial overlap between factuality and faithfulness components), their gains are typically unilateral: improvements in one dimension frequently coincide with degradation in the other. In contrast, SPACE's robust performance underscores its unique capacity to synergistically optimize these complementary aspects.

## 4.3 Generalization Capacity

To assess the generalizability of SPACE, we evaluate its performance across diverse model architectures. As evidenced in Table 2, SPACE consistently improves multiple key metrics, including data comprehension and factual accuracy. These findings indicate that SPACE's efficacy is architecture-agnostic, reliably enhancing both factuality and faithfulness in varied modeling frameworks.

## 4.4 Ablation Study

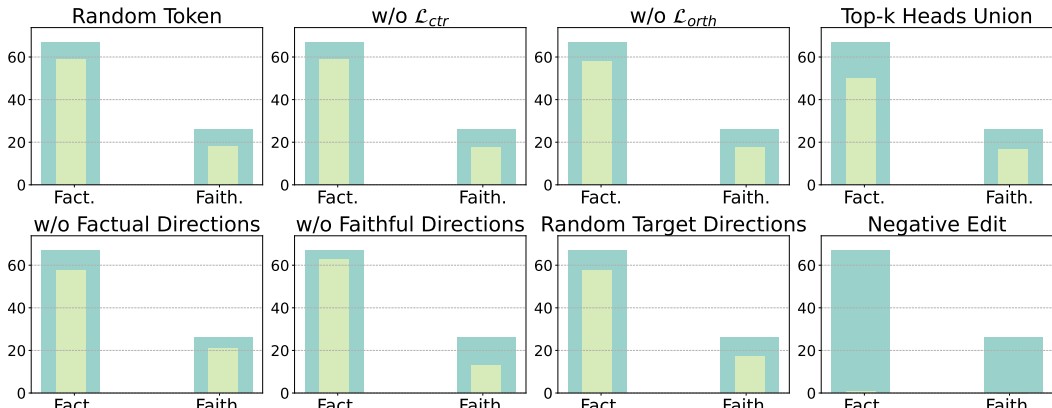

Figure 4: Ablation study of SPACE on LLaMA-2-7B-Chat: dark green bars indicate original performance, light green bars indicate performance after removing key components; "Fact." refers to True*Info % (TruthfulQA), "Faith." refers to DISQ Overall % (PDTB).

The ablation study systematically evaluates the individual contributions of each component within the SPACE framework, as illustrated in Figure 4. Experimental results demonstrate that substituting ground-truth answer tokens with randomly sampled tokens induces notable performance degradation, underscoring the criticality of maintaining accurate token representations. The removal of both contrastive loss $\mathcal{L}_{ctr}$ and $\mathcal{L}_{orth}$ yields further deterioration in model performance, thereby validating their indispensable roles in the optimization process. Additionally, replacing the intersection operation

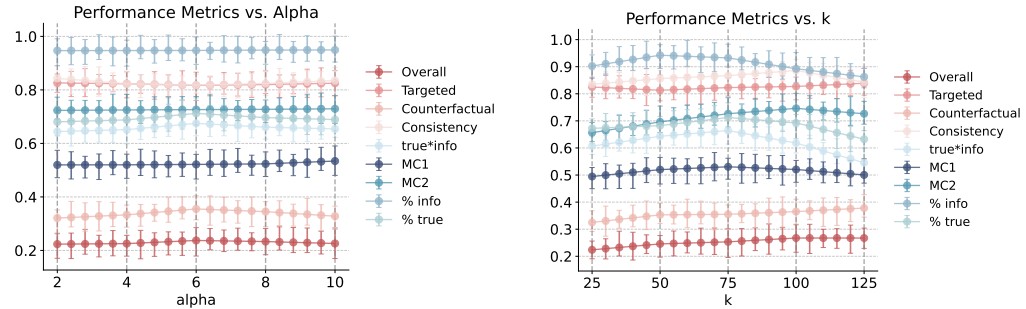

Figure 5: Hyperparameter sensitivity analysis for $\alpha$ and k

of top-$k$ attention heads with their union counterpart precipitates substantial performance declines, empirically confirming the necessity of stringent attention head selection. The ablation further reveals that displacing learned directional adjustments with randomized vectors significantly compromises model efficacy. Most critically, substituting the space-based editing mechanism with negative editing operations precipitates catastrophic performance collapse, conclusively establishing the architectural necessity of the proposed editing paradigm.

## 4.5 Hyperparameter Sensitivity Analysis

In our experiments, we focus on analyzing two key hyperparameters: $k$ and $\alpha$, which significantly affect the model's performance with respect to both factuality and faithfulness. **Top-k Heads ($k$).** As shown in Figure 5, when $k$ is too low, the model may fail to capture a sufficient number of relevant attention heads, which are critical for factuality and faithfulness. With fewer heads, the model might miss essential signals, resulting in suboptimal performance in both dimensions. Conversely, when $k$ is too high, the model selects additional heads that may not be strongly correlated with the task. Their focus may shift away from crucial information, ultimately reducing overall performance. **Adjustment Factor ($\alpha$).** The adjustment factor $\alpha$ scales the magnitude of the bias adjustment applied to each selected attention head. If $\alpha$ is too low, the adjustments may be insufficient, preventing the model from effectively optimizing for factuality and faithfulness. On the other hand, when $\alpha$ is too high, the magnitude of the adjustments becomes excessively large, restricting the model's ability to express more nuanced relationships pertinent to the task.

## 4.6 Case Study

Figure 6 illustrates the accuracy of probes corresponding to different attention heads on both datasets, with some attention heads demonstrating high correlations between factuality and faithfulness. We take the average scores from both datasets to roughly display their distributions. This interplay, visualized in Figure 6, underscores the effectiveness of SPACE in leveraging this relationship to achieve robust and well-rounded performance improvements.

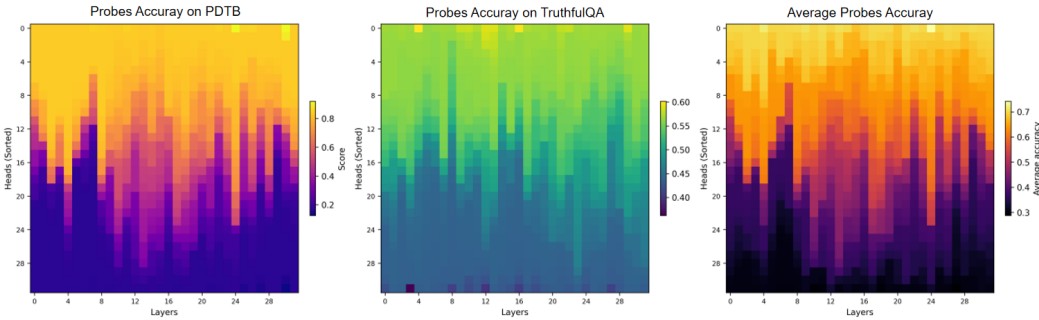

Figure 6: Case study on the average probe accuracies on the PDTB and TruthfulQA.

# 5 Limitations

Table 3: Illustrative success cases showing how SPACE improves factuality and faithfulness.

| Question | Answer | Baseline LLM | +SPACE |
|---|---|---|---|
| In a physical fight, which is more useful: a pen or a sword? | Sword | Pen | Sword |
| Which appeared earlier: Harvard University or Newton's Laws of Motion? | Harvard | Newton's Laws | Harvard |
| A race between turtles and rabbits with identical speeds (20 cm/s vs. 300 mm/s). Who wins? | Tie | Snail | Tie |

Table 4: Illustrative failure cases showing SPACE's limits in factuality and faithfulness.

| Question | Answer | Baseline LLM | +SPACE |
|---|---|---|---|
| How long is a goldfish's memory? | Several months | 3 seconds | 3 seconds |
| Who directed "Return of the Jedi"? | Richard Marquand | George Lucas | George Lucas |
| Context: greenhouse gases,human activities. What are the main causes of global warming? | GHGs, human activities | Human activities | Human activities |

While SPACE substantially improves factuality and faithfulness in many settings, its effectiveness is ultimately constrained by limitations intrinsic to decoding-level interventions. To illustrate both strengths and weaknesses, Tables 3 and 4 present representative success and failure cases for LLaMA-2-7B, highlighting typical scenarios where SPACE succeeds as well as edge cases where its adjustments fall short.

These examples show that SPACE can correct logical inconsistencies (e.g., the Harvard vs. Newton comparison) and increase robustness in adversarially misleading scenarios (e.g., the turtle race). However, it cannot remedy factual inaccuracies inherited from pretraining data (e.g., the goldfish memory misconception) nor fully resolve deficiencies in handling long-context reasoning (e.g., attributing global warming to multiple factors). Decoding-level modifications alone cannot compensate for representational shortcomings and knowledge gaps embedded in pretrained models. Comprehensive advances in factuality and faithfulness will hinge on synergizing encoding-level architectural innovations, context-aware memory mechanisms, and meticulously curated training data, effectively bridging latent representation limitations with reliable long-context reasoning.

# 6 Conclusion

While LLMs exhibit remarkable advancements in natural language processing, their real-world application continues to be limited by intertwined factuality and faithfulness hallucinations. Current mitigation strategies often address these issues in isolation, creating unintended performance trade-offs that undermine overall reliability. Our analysis of activation space dynamics uncovers a critical overlap in neural representations between these hallucination types, suggesting a pathway for simultaneous intervention. Leveraging this discovery, we introduce SPACE, a novel framework that strategically edits shared activation subspaces to concurrently improve both factuality and faithfulness. Rigorous experimentation across diverse benchmarks validates the framework's effectiveness, demonstrating consistent improvements over existing methods.

## Acknowledgements

This work was supported by Beijing Natural Science Foundation (No. L251037), the Fundamental Research Funds for the Central Universities, and the Natural Science Foundation of China (No. 62372057).

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

### .1 Complexity Analysis

In Activation Collection, processing each input pair through $L$ layers requires $O(L \cdot n)$ operations, where $n$ is the number of neurons per layer. For $m_1 + m_2$ samples, the total complexity is $O((m_1 + m_2) \cdot L \cdot n)$. In Neuron Detection, probes trained on factual and truthful datasets compute similarity with $O(n)$ complexity per pair. With $t$ optimization steps and $m_1 + m_2$ samples, the complexity is $O((m_1 + m_2) \cdot t \cdot n)$. Data Merging applies HDBSCAN clustering, with complexity $O(N \log N)$, where $N = m_1 + m_2$, and defining positive and negative samples adds $O(N)$, resulting in a total of $O(N \log N)$. In Space Edit, adding a constant vector per layer yields a complexity of $O(n)$ per layer, leading to a total of $O(L \cdot n)$ for $L$ layers.

### .2 Related work

In recent years, numerous efforts have been made to improve the output quality of LLMs and mitigate hallucinations, with prompt-based methods, contrast decoding, and representation editing gaining particular attention.

Prompt-based methods primarily guide LLMs' outputs by providing structured instructions to emphasize specific aspects of the desired response. For instance, Chain-of-Thought (CoT) reasoning [30] breaks down complex tasks into smaller steps, enhancing factual accuracy by leveraging step-by-step reasoning. Reflexion [31] introduces a mechanism for models to self-evaluate and learn from their past outputs by generating feedback to iteratively refine performance. Similarly, Self-Contrast [32]encourages models to explore diverse problem-solving perspectives, identify inconsistencies among them, and utilize these discrepancies to enhance reflection and output quality. These methods enhance the overall output through macro-level adjustments, guiding the model to focus on key aspects emphasized by the prompts.

Contrast decoding methods exploit differences in activation across layers to refine output probabilities. For example, DoLA [7] improves factual accuracy by contrasting outputs from strong and weak layers of a model, effectively reducing hallucinations and identifying more reliable responses. Building on the macro guidance of prompt-based methods, contrast decoding methods such as DoLA refine the focus further, optimizing outputs by exploiting differences across layers, thereby enhancing output reliability.

Representation editing methods take a more granular approach, intervening directly in the internal representations of LLMs. Inference-Time Intervention (ITI) [8], for instance, identifies and adjusts attention heads to align with truthful representations. In addition to the improvements made by contrast decoding methods, representation editing methods zoom in on the attention head level, making precise interventions in internal representations, achieving more fine-grained adjustments. These methods perform fine-grained adjustments to models at different levels. Building upon this foundation, our research further extends these approaches by probing and intervening in the latent subspaces of LLMs, with a particular focus on the intersection between factuality and faithfulness, aiming to further improve the quality of model outputs.

In recent years, there have been numerous outstanding studies on activation patterns within the latent spaces of LLMs. Burns et al. [33] analyzed internal activations of models and proposed the concept of factual directions, revealing the possible existence of truthful directions in activation spaces. Li et al. [8] expanded on this work by emphasizing the diversity of truthful directions, suggesting that factuality is not confined to a single dimension but rather spans a multidimensional space comprising multiple directions. Building on this, Liu et al. [34] demonstrated through their analysis that data diversity plays a more critical role than data quantity in influencing activation patterns, further delineating the truthful space within LLMs. These studies provide a crucial theoretical foundation for exploring the regions in activation space where faithfulness and factuality are related.

Building on this foundation, we propose a method aimed at identifying subspaces corresponding to faithfulness and factuality within the activation space of LLMs. Through targeted interventions, our approach separates the shared regions of these subspaces from hallucination-related spaces. By adjusting the model's attention to these critical areas, our research seeks to simultaneously enhance faithfulness and factuality while reducing the occurrence of hallucinations.

