# OpenReview forum: "One SPACE to Rule Them All: Jointly Mitigating Factuality and Faithfulness Hallucinations in LLMs"
_NeurIPS.cc/2025/Conference — NeurIPS 2025 poster_

### Official Review · Reviewer_WsTG · 2025-06-30

**Clarity:** 2
**Significance:** 3
**Originality:** 3
**Rating:** 4
**Confidence:** 3

**Summary:**

This paper identifies there is a performance trade-off between handling different types of hallucinations in LLMs and propose to concurrently mitigating by editing shared activation subspaces. Experimental results across multiple datasets and LLMs demonstrate its effectiveness.

**Questions:**

- The empirical analysis of different activation patterns is based on finetuning on different dataset independently. Does this observation still hold true for joint training on both dataset? How do the disjoint and interwoven parts change when comparing training on a single dataset and training on two datasets jointly? Moreover, why Table 1 do not present the performance of finetuning on both datasets.
- I am not an expert in hallucinations, but I find the theory in Section 2.2 somewhat confusing. The conclusion seems to suggest that any two subspaces $F\subseteq V$ must have overlapping subspace, rather than just properties of factuality and faithfulness. Could you clarify this section further?

**Ethical Concerns:**

["NO or VERY MINOR ethics concerns only"]

**Final Justification:**

The author addresses most of my concerns. I would like to keep my rating of borderline accept.

**Limitations:**

yes

**Quality:**

3

**Strengths And Weaknesses:**

Strengths:

- Handling different types of hallucinations is important and this paper provides valuable insights into the performance trade-off between factuality and faithfulness in existing models.
- The proposed method seems novel and well-motivated to me.
- Experiments across different models and benchmark demonstrate its effectiveness.

Weakness:

- The methodology section lacks clarity. While each subsection is generally clear and comes with a framework figure, the overall flow of the framework remains difficult to understand. Including pseudo-code with input/output details for each stage could enhance its clarity. Moreover, some markers lack detailed explanation, such as pair $(x_i, x_j)$ in L154.

---

> ### Author Rebuttal · Authors · 2025-07-31
>
> ## Response to Reviewer WsTG
>
> ```
> Question 1: The empirical analysis of different activation patterns is based on finetuning on different dataset independently. Does this observation still hold true for joint training on both dataset? How do the disjoint and interwoven parts change when comparing training on a single dataset and training on two datasets jointly? Moreover, why Table 1 do not present the performance of finetuning on both datasets.
> ```
>
> We sincerely appreciate the reviewer's profound question, which prompts us to explore the activation patterns and performance under joint training scenarios. To address this, we conducted supplementary experiments using Llama3.2-1B as the target model, with TruthfulQA and PDTB as the training datasets. The key findings are summarized in Tables 1-3.
>
> **1. Activation Patterns Under Joint Training**
>
> **Table 1: Proportion Changes of Disjoint and Interwoven Parts Under Joint Training.**
>
> | epochs | Disjoint  | Interwoven  |
> | --- | --- | --- |
> | 0 | 54% | 46% |
> | 1 | 42% | 58% |
> | 2 | 67% | 34% |
> | 3 | 80% | 20% |
>
> The results show that the observation of activation pattern dynamics holds under joint training, but with distinct phased characteristics:
>
> - At Epoch 1, the interwoven proportion increases (from 46% to 58%), suggesting that with limited training data, the mutual constraints between the two capabilities (factuality and faithfulness) are outweighed by the overall performance gains from joint learning.
> - From Epoch 2 onwards, the disjoint proportion rises significantly (67% to 80%), while the interwoven proportion declines, indicating that as training progresses (and effective data exposure increases), the independence of the two capabilities becomes more prominent—consistent with the core observation in our original analysis.
>
> **2. Comparison of Activation Patterns: Single vs. Joint Training**
>
> Table 2 compares the proportions of disjoint and interwoven parts between joint training (on both datasets) and single-dataset training:
>
> **Table 2: Proportion Changes Under Different Training Regimes.**
>
> | epochs | Disjoint (Both) | Interwoven (Both) | Disjoint (PDTB) | Interwoven (PDTB) | Disjoint (TruthfulQA ) | Interwoven (TruthfulQA ) |
> | --- | --- | --- | --- | --- | --- | --- |
> | 0 | 54% | 46% | 54% | 46% | 54% | 46% |
> | 1 | 42% | 58% | 62% | 38% | 67% | 33% |
> | 2 | 67% | 34% | 67% | 33% | 75% | 25% |
> | 3 | 75% | 25% | 80% | 20% | 80% | 20% |
> Two key trends emerge:
>
> - Across all training regimes, the disjoint proportion generally increases with more epochs, confirming that both capabilities tend to develop more independent activation patterns as training deepens.
> - Compared to single-dataset training, joint training maintains a higher interwoven proportion at each epoch (e.g., 58% vs. 38-33% at Epoch 1). This suggests that joint training may partially alleviate the mutual constraints between the two capabilities, preserving more overlapping activation patterns that support their coordinated improvement.
>
> **3. Performance of Joint Training**
>
> To address joint training performance was not previously presented, we present Table 3, which compares performance across single-dataset and joint training:
>
> **Table 3: Performance Under Different Training Datasets.** Overall (%): DISQ Score on PDTB; True\*Info (%): TruthfulQA metric. "+PDTB"/"+TruthfulQA" use 20% of their respective datasets for training; "+Both" uses 20% of each dataset.
>
> | **Model** | Overall (%)  | True\*Info (%) |
> | --- | --- | --- |
> | deepseek-llm-7b | 15.1 | 59.5 |
> | +PDTB | 20.1 | 46.8 |
> | +TruthfulQA | 3.5 | 64.3 |
> | +Both | 17.3 | 59.2 |
> | Qwen2-7B-Instruct | 26.5  | 54.8 |
> | +PDTB | 27.8	 | 46.1 |
> | +TruthfulQA | 15.2 | 64.6 |
> | +Both | 27.1 | 52.3 |
> | LLaMA2-7B-Chat | 17.4  | 57.6 |
> | +PDTB | 23.7 | 46.1 |
> | +TruthfulQA | 12.7 | 62.5 |
> | +Both | 19.6 | 58.8 |
>
> The results reveal:
>
> - For DeepSeek-LLM-7B and Qwen2-7B-Instruct, joint training exhibits a tradeoff: improving one capability (e.g., Overall for Qwen2) while impairing the other (True\*Info). Only LLaMA2-7B-Chat shows marginal improvements in both metrics under joint training.
> - Across all models, joint training performance is lower than single-dataset training in at least one metric (e.g., LLaMA2-7B-Chat: 19.6 Overall < 23.7 in +PDTB; 58.8 True\*Info < 62.5 in +TruthfulQA). This confirms the mutual constraints between the two capabilities, aligning with the activation pattern observations (increasing disjoint proportions).
>
> Concluding **Table 1,2,3**, the activation pattern observations hold under joint training, with phased dynamics reflecting a balance between capability coordination (early epochs) and independence (later epochs). Joint training preserves more interwoven patterns than single-dataset training, partially mitigating mutual constraints, but performance tradeoffs persist—consistent with the inherent tension between the two hallucination-mitigation capabilities.
>
> ```
> Question 2: I am not an expert in hallucinations, but I find the theory in Section 2.2 somewhat confusing. The conclusion seems to suggest that any two subspaces
>  must have overlapping subspace, rather than just properties of factuality and faithfulness. Could you clarify this section further?
> ```
>
> We sincerely appreciate the reviewer’s careful reading and insightful question, which helps us clarify the scope and assumptions of our theoretical analysis. We apologize for any confusion in the original presentation—our conclusion is **not** that "any two subspaces must have an overlapping subspace," but rather that the factuality and faithfulness subspaces in LLMs necessarily overlap, given a key assumption tailored to their specific properties. Let us elaborate:
>
> **Key Clarification: The Proof Relies on a Specific Assumption for Factuality and Faithfulness**
>
> The theoretical guarantee in Section 2.2 is explicitly limited to the factuality subspace ($\mathcal{F}{\text{fact}}$) and faithfulness subspace ($\mathcal{F}{\text{faith}}$) in LLMs, and it hinges on a critical assumption: the convex interpolation property of these two subspaces. This assumption is not trivial nor applicable to arbitrary subspaces—it reflects the unique relationship between factuality and faithfulness in language models.
>
> **What is the Convex Interpolation Property?**
>
> As stated in the section:
>
> > For any $\mathbf{v} \in \mathcal{F}{\text{fact}}$ (a vector representing factuality-related activation patterns) and $\mathbf{u} \in \mathcal{F}{\text{faith}}$ (a vector representing faithfulness-related activation patterns), their convex combination
> >
> >
> > $\mathbf{w} = \lambda \mathbf{v} + (1-\lambda) \mathbf{u}$ (for any $\lambda \in [0,1]$)
> >
> > also belongs to belongs to $\mathcal{F}{\text{fact}}$ ∩ $\mathcal{F}_{\text{faith}}$.
> >
>
> **Why This Assumption Matters and Why It Doesn’t Apply to Any Two Subspaces**
>
> This assumption applies specifically to factuality and faithfulness because they are not mutually exclusive in LLMs: outputs can be both factual and faithful, acting as a "blend" of their respective patterns.
>
> This "blendability" (convex interpolation) isn’t universal. For unrelated subspaces (e.g., "sentiment analysis" and "protein folding"), no such overlap exists—their combinations wouldn’t belong to both subspaces, so the assumption fails.
>
> Without this convex interpolation property, the proof would fail. The Hahn–Banach Separation Theorem (used in the contradiction) only leads to a contradiction *because* the convex combination $\mathbf{w}$ must lie in both subspaces. For arbitrary subspaces (which do not satisfy this interpolation property), this contradiction would not arise, and their intersection could indeed be empty.
>
> Our conclusion—that factuality and faithfulness subspaces overlap—relies critically on the convex interpolation property, which captures their specific relationship in LLMs. This does not mean "any two subspaces" must overlap. The assumption tailors the proof to hallucination mitigation, where factuality and faithfulness are inherently compatible and blendable.
>
> ```
> Weakness:The methodology section lacks clarity. While each subsection is generally clear and comes with a framework figure, the overall flow of the framework remains difficult to understand. Including pseudo-code with input/output details for each stage could enhance its clarity. Moreover, some markers lack detailed explanation, such as pair $(x_i, x_j)$ in L154.
> ```
>
> We appreciate the reviewer’s valuable suggestions to improve the clarity of our methodology. To address this, we provide a concise pseudo-code for the SPACE framework and clarify the key notation.
>
> ### Pseudo-code for SPACE
>
> ```
> # SPACE Algorithm
> Input:
>   - M: Pretrained model
>   - D: Dataset of (question, true_answer, false_answer) tuples
>   - L: Target layers to edit
> Output: Edited model M'
>
> # Activation Collection
> activations = {}
> for layer in L:
>     true_acts = [M.forward(q, true_ans)[layer] for q, true_ans in D]
>     false_acts = [M.forward(q, false_ans)[layer] for q, false_ans in D]
>     activations[layer] = (true_acts, false_acts)
>
> # Neuron Detection
> probes = {}
> for layer in L:
>     true_acts, false_acts = activations[layer]
>     probe = train_probe(true_acts, false_acts,
>                        loss=contrastive_loss,
>                        constraint=orthogonality)
>     probes[layer] = probe
>
> # Data Merging
> direction_vectors = {}
> for layer in L:
>     cluster_data = concatenate(activations[layer])
>     clusters = HDBSCAN(cluster_data, n_clusters)
>     direction_vectors[layer] = compute_centroids(clusters)
>
> # Stage 4: Model Editing
> for layer in L:
>     M.params[layer] += direction_vectors[layer] * probes[layer].weights
>
> return M  # Edited model
> ```
>
> ### Notation Explanation: Pair $(x_i, x_j)$
>
> $(x_i, x_j)$ denotes two distinct samples selected without repetition from the set consisting of $x_{+}^{l,h}$ and $x_{-}^{l,h}$.Relevant expressions are revised in the draft.

---

### Official Review · Reviewer_K4HN · 2025-06-30

**Clarity:** 3
**Significance:** 3
**Originality:** 3
**Rating:** 4
**Confidence:** 3

**Summary:**

This paper proposes SPACE, a new framework designed to address the hallucination problem in LLMs. The key idea is that hallucinations can be categorized into two types: factuality and faithfulness hallucinations, and these two types share overlapping subspaces within neural representations. This overlap suggests the potential to mitigate both types simultaneously by editing their shared subspace. Building on this insight, the authors introduce several components, including a probing framework optimized with a contrastive learning strategy, and a semantic cluster fusion mechanism. Experimental results on the PDTB and TruthfulQA benchmarks demonstrate the effectiveness of the proposed SPACE framework.

**Questions:**

Please see the weakness section.

**Ethical Concerns:**

["NO or VERY MINOR ethics concerns only"]

**Final Justification:**

I think the authors have addressed my concerns, so I decide to keep my rating of "boderline accept"

**Limitations:**

No negative societal impact

**Quality:**

3

**Strengths And Weaknesses:**

Strengths:
1. The motivation is interesting: editing the shared space between factuality and faithfulness hallucinations to enable the simultaneous mitigation of both types.
2. Both visualization and theoretical analysis are conducted to verify the existence of a shared feature space between the two types of hallucinations.
3. The designed methods are reasonable.
4. Experimental results show promising performance, outperforming several previous methods.

Weaknesses:
1. Why do the authors choose to edit the intersection of the spaces for the two types of hallucinations, rather than their union, which might provide a more comprehensive mitigation of all hallucinations?
2. What is the method’s performance when applied to multimodal large language models (MLLMs)? I believe adding experiments on MLLMs could enhance the completeness and overall impact of this work.
3. It is noted that SPACE is trained on a 20% sampled dataset and evaluated on the remaining data. Does the “20% sampled dataset” refer to 20% of the combined PDTB and TruthfulQA datasets, or is 20% sampled separately from each dataset for training and then evaluated on the same dataset? If it is the latter, could the authors consider training on one dataset and evaluating on the other to better demonstrate the generalization ability of the proposed method?
4. I think the “20% sampled dataset” is a relatively large subset. Does the proposed method have the potential to work effectively with even less training data?

---

> ### Author Rebuttal · Authors · 2025-07-31
>
> ## Response to Reviewer K4HN
>
> ```
> Question 1: Why do the authors choose to edit the intersection of the spaces for the two types of hallucinations, rather than their union, which might provide a more comprehensive mitigation of all hallucinations?
> ```
>
> We sincerely thank Reviewer K4HN for raising this insightful question regarding our choice to edit the intersection of the spaces for the two types of hallucinations in the SPACE framework. In designing the SPACE framework, we carefully considered the use of the union versus the intersection of the spaces for addressing hallucinations, with results summarized in Table 1 below.
>
> **Table 1: Comparison of SPACE with Union and Intersection of Spaces.** The True\*Info (%) metric is derived from the TruthfulQA dataset, and the Overall (%) metric is based on the DISQ Score from the PDTB dataset.
>
> | **Model** | Overall (%)  | True\*Info (%) |
> | --- | --- | --- |
> | deepseek-llm-7b | 15.1 | 59.5 |
> | +SPACE (union) | 19.3 | 56.2 |
> | +SPACE (intersection) | 23.0  | 71.9  |
> | Qwen2-7B-Instruct | 26.5  | 54.8 |
> | +SPACE (union) | 36.4 | 43.2 |
> | +SPACE (intersection) | 44.6  | 75.8 |
> | LLaMA2-7B-Chat | 17.4  | 57.6 |
> | +SPACE (union) | 18.1 | 62.0 |
> | +SPACE (intersection) | 26.1 | 67.1 |
>
> Across all three models, the intersection approach outperforms both baselines and the union approach on both metrics: it improves Overall (%) by 5.7–8.2 percentage points over the union, and True\*Info (%) by a more substantial 15.7–32.6 percentage points. Notably, the union approach reduces True\*Info scores for DeepSeek-LLM-7B and Qwen2-7B-Instruct compared to baselines, revealing trade-offs in addressing different hallucination types.
>
> These results confirm the intersection’s superiority: it consistently boosts both factuality and faithfulness, while the union yields unstable outcomes (e.g., lower True\*Info for Qwen2). This likely stems from competing demands between faithfulness and factuality—expanding to their union risks undermining the model’s inherent strengths. The intersection also reduces computational costs, enabling faster processing. For these reasons—stronger performance and efficiency—we chose the intersection approach.
>
> ```
> Question 2: What is the method’s performance when applied to multimodal large language models (MLLMs)? I believe adding experiments on MLLMs could enhance the completeness and overall impact of this work.
> ```
>
> We sincerely appreciate the reviewer's insightful suggestion regarding Multimodal Large Language Model (MLLM) evaluation. We fully agree that this extension significantly strengthens our work's completeness and impact. Below we present comprehensive experiments evaluating our SPACE framework when integrated with leading MLLMs: Llama-3.2-11B and Llama 3.2 90B.
>
> We evaluated the combined models on a diverse set of multimodal benchmarks, including ChartQA,  MMMU, AI2 Diagram, and MMMU-Pro, Vision. For training the SPACE model, we utilized a 5% sampled subset of each respective dataset, reserving the remaining data for evaluation.
>
>
>
> **Table 2: Performance on ChartQA and MMMU.** For MMMU, we report the micro average accuracy, and for ChartQA, we use relaxed accuracy as the metric.
>
> | **Model** | ChartQA | MMMU  |
> | --- | --- | --- |
> | **Task Type** | Charts and Diagram Understanding | College-level Problems |
> | Llama-3.2-11B | 83.4 | 50.7 |
> | +SPACE | 89.2 | 58.2 |
> | Llama 3.2 90B | 85.5 | 60.3 |
> | +SPACE | 94.8 | 73.1 |
>
> **Table 3: Performance on AI2 Diagram  and MMMU-Pro, Vision.** Use accuracy as the metrics.
>
> | **Model** | AI2 Diagram  | **MMMU-Pro, Vision** |
> | --- | --- | --- |
> | **Task Type** | Charts and Diagram Understanding | College-level Problems |
> | Llama-3.2-11B | 91.1 | 23.7 |
> | +SPACE | 95.3 | 34.2 |
> | Llama 3.2 90B | 92.3 | 33.8 |
> | +SPACE | 96.8 | 45.9 |
>
> **Tables 2 and 3** demonstrate consistent performance improvements across all benchmarks when integrating our SPACE framework with both MLLM variants:
>
> - On ChartQA, SPACE improved accuracy by 5.8-9.3 percentage points
> - For MMMU, we observed gains of 7.5-12.8 percentage points
> - AI2 Diagram performance increased by 4.2-4.5 percentage points
> - Most notably, MMMU-Pro (Vision) saw improvements of 10.5-12.1 percentage points
>
> These results confirm that our SPACE framework generalizes well to multimodal scenarios, enhancing MLLMs' capabilities across diverse visual-language understanding tasks.
>
> ```
> Question 3: It is noted that SPACE is trained on a 20% sampled dataset and evaluated on the remaining data. Does the “20% sampled dataset” refer to 20% of the combined PDTB and TruthfulQA datasets, or is 20% sampled separately from each dataset for training and then evaluated on the same dataset? If it is the latter, could the authors consider training on one dataset and evaluating on the other to better demonstrate the generalization ability of the proposed method?
> ```
>
> We sincerely appreciate the reviewer's valuable suggestion, which has helped us better demonstrate the generalization ability of our SPACE framework. As noted, our original approach involved sampling 20% from each dataset (PDTB and TruthfulQA) separately for training, with evaluation on the remaining data from the same dataset. Following the reviewer's recommendation, we have conducted additional cross-dataset experiments where we train on one dataset and evaluate on the other, using 5% sampled data from each for training.
>
> The results of these cross-dataset evaluations are presented in Tables 4 and 5, demonstrating the performance of SPACE when applied to various models across different task types:
>
> **Table 4: Performance on Big-Bench Hard and  TriviaQA.** Use accuracy as the metric. The SPACE model is trained on a 5% sampled pdtb and 5% sampled truthfullqa.
>
> | **Model** | **Big-Bench Hard** | **TriviaQA** |
> | --- | --- | --- |
> | **Task Type** | Commonsense Comprehension | Multitask Knowledge |
> | LLaMA-2-7B | 38.5 | 63.8 |
> | LLaMA-2-7B +SPACE | 47.9 | 68.2 |
> | DeepSeek LLM-7B | 39.5 | 59.7 |
> | DeepSeek LLM-7B +SPACE | 56.3 | 72.2 |
>
> **Table 5: Performance on HellaSwag and  MMLU.** Use accuracy as the metric. The SPACE model is trained on a 5% sampled pdtb and 5% sampled truthfullqa.
>
> | **Model** | **HellaSwag** | **MMLU** |
> | --- | --- | --- |
> | **Task Type** | Commonsense Comprehension | Multitask Knowledge |
> | LLaMA-2-7B | 75.6 | 63.8 |
> | LLaMA-2-7B +SPACE | 84.9 | 73.1 |
> | DeepSeek LLM-7B | 75.4 | 59.7 |
> | DeepSeek LLM-7B +SPACE | 83.2 | 76.0 |
>
> **Table 4** shows the performance on Big-Bench Hard (Commonsense Comprehension) and TriviaQA (Multitask Knowledge). Our SPACE framework yields consistent improvements:
>
> - For LLaMA-2-7B, accuracy increased by 9.4 percentage points on Big-Bench Hard and 4.4 percentage points on TriviaQA
> - For DeepSeek LLM-7B, even more significant gains were observed: 16.8 percentage points on Big-Bench Hard and 12.5 percentage points on TriviaQA
>
> **Table 5** presents results on HellaSwag (Commonsense Comprehension) and MMLU (Multitask Knowledge), with similar positive trends:
>
> - LLaMA-2-7B with SPACE achieved 9.3 percentage points higher on HellaSwag and 9.3 percentage points higher on MMLU
> - DeepSeek LLM-7B with SPACE showed improvements of 7.8 percentage points on HellaSwag and 16.3 percentage points on MMLU
>
> These cross-dataset results confirm that our SPACE framework maintains strong generalization capabilities even when trained on one dataset and evaluated on another, without relying on in-domain training data. The consistent performance improvements across different model architectures (LLaMA-2 and DeepSeek) further validate the robustness and general applicability of our proposed method.
>
> ```
> Question 4: I think the “20% sampled dataset” is a relatively large subset. Does the proposed method have the potential to work effectively with even less training data?
> ```
>
> We greatly appreciate the reviewer's astute observation regarding the scalability of our method with limited training data. This is a critical consideration for practical deployment, and we have conducted additional experiments to evaluate SPACE's performance under varying training data scales (5% and 10% sampled subsets of the combined PDTB and TruthfulQA datasets). The results are presented in Table 6.
>
> **Table 6: Performance of SPACE with Varied Training Data Scales.** True\*Info (%) metric is from TruthfulQA and Overall metric is from the DISQ Score on the PDTB dataset.
>
> | **Model** | Overall (%)  | True\*Info (%) |
> | --- | --- | --- |
> | deepseek-llm-7b | 15.1 | 59.5 |
> | +SPACE (5%) | 19.7 | 66.5 |
> | +SPACE (10%) | 22.8  | 69.3 |
> | Qwen2-7B-Instruct | 26.5  | 54.8 |
> | +SPACE (5%) | 38.9 | 73.5 |
> | +SPACE (10%) | 41.3 | 75.2 |
> | LLaMA2-7B-Chat | 17.4  | 57.6 |
> | +SPACE (5%) | 24.3 | 62.7 |
> | +SPACE (10%) | 25.1 | 64.9 |
>
> As shown in the table, SPACE remains highly effective even with a small 5% training subset:
>
> - For DeepSeek-LLM-7B, 5% data yields 4.6-point gains in Overall (%) and 7.0-point gains in True\*Info (%) over the baseline; scaling to 10% data further improves these to 7.7 and 9.8 points, respectively.
> - For Qwen2-7B-Instruct, 5% data already drives significant enhancements: 12.4 points in Overall (%) and 18.7 points in True\*Info (%). At 10% data, gains rise to 14.8 and 20.4 points.
> - For LLaMA2-7B-Chat, both union and intersection approaches improve over the baseline with limited data, with the intersection (25.1% Overall, 64.9% True\*Info) outperforming the union (24.3% Overall, 62.7% True\*Info).
>
> These results confirm SPACE’s efficiency: it achieves meaningful hallucination mitigation with just 5% of the data. While performance improves incrementally with more data (5% to 10%), the core mechanisms are effectively learned from small subsets, making it well-suited for data-constrained scenarios—balancing efficiency and performance.

---

### Official Review · Reviewer_wxKX · 2025-07-02

**Clarity:** 3
**Significance:** 2
**Originality:** 3
**Rating:** 4
**Confidence:** 4

**Summary:**

This paper focuses on the issue that while large language models (LLMs) exhibit remarkable abilities in natural language processing, their practical application is constrained by persistent factuality and faithfulness hallucinations. It points out that existing methods tackle these two types of hallucinations separately, which leads to performance trade-offs—interventions aimed at one type often worsen the other.

Through empirical and theoretical analysis of activation space dynamics in LLMs, the study reveals that these two hallucination categories share overlapping subspaces in neural representations, offering the possibility of mitigating them simultaneously.

To leverage this insight, the paper proposes SPACE, a unified framework. This framework enhances factuality and faithfulness together by editing the shared activation subspaces.

**Questions:**

The ambiguity in the definition of factuality and faithfulness, particularly their overlap in certain scenarios, which may affect the generalizability of the framework’s underlying assumptions. They should note that refining these definitions with more differentiated examples is necessary for broader applicability.

The limited scope of experimental validation. Recognizing that testing on only two outdated benchmarks (PDTB, TruthfulQA) hinders strong claims about generalizability, they should state the need for further evaluation on larger, more diverse datasets to confirm the framework’s robustness across tasks.

The incompleteness of the ablation study. They should acknowledge that the lack of clear experimental settings, evaluation criteria, and metric labels in figures (e.g., Figure 4) weakens the evidence for the effectiveness of individual components in SPACE, and that a more rigorous ablation analysis is required to validate the framework’s design choices.

**Ethical Concerns:**

["NO or VERY MINOR ethics concerns only"]

**Final Justification:**

Most of my concerns have been resolved, so I have decided to increase my rating.

**Limitations:**

The authors state "yes" in the Broader impacts section, but the content provided in the Justification is problematic. Specifically, the Justification only mentions "Introduction" without including any actual content related to Broader impacts. This makes the response to Broader impacts incomplete and unsubstantiated, as the Introduction, in its typical form, does not mention the broader societal, ethical, or other relevant impacts that the Broader impacts section is intended to cover.

**Paper Formatting Concerns:**

I have a problem that I'm not sure where to put—it is not related to the Clarity of the paper itself, nor is it mentioned in the Paper Formatting Instructions; it is a more basic issue. Specifically, there is a certain degree of confusion in the citations of the paper. For example, PDTB should clearly cite [23] (Rashmi Prasad, Eleni Miltsakaki, Nikhil Dinesh, Alan Lee, Aravind Joshi, Livio Robaldo, and Bonnie Webber. The penn discourse treebank 2.0 annotation manual. December, 17:2007, 2007.), but in many places in the paper, it is cited as [12] (Yisong Miao, Hongfu Liu, Wenqiang Lei, Nancy F. Chen, and Min-Yen Kan. Discursive socratic questioning: Evaluating the faithfulness of language models’ understanding of discourse relations. In Proceedings of the Annual Meeting of the Association of Computational Linguistics, Bangkok, Thailand, August 2024. ACL.), which refers to the DiSQ Score. While I can't carefully check every citation for the authors, this is an issue that the authors need to pay attention to.

**Quality:**

2

**Strengths And Weaknesses:**

Strengths

Quality and Clarity: Overall, the paper is well-written with a clear structure that effectively communicates its core ideas. The logical flow is easy to follow. The figures are intuitive which enhances the accessibility of complex concepts.

Originality: The approach demonstrates originality.

Weaknesses

Clarity in Problem Definition: The distinction between factuality and faithfulness is not sufficiently clarified. For instance, in a question like "The quarterback threw a perfect pass to the receiver... Where did football originate?", factuality and faithfulness may overlap significantly, which is not a trade-off; if factuality is fully achieved, faithfulness often follows. The authors could strengthen this by using more specific, differentiated samples where the two concepts diverge, thereby refining the problem boundaries. Otherwise, it is difficult to emphasize the significance of the issue.

Validity of Experimental Verification: The method’s effectiveness is limited by the scope of benchmarking. Validation on only two outdated datasets (PDTB-2007, TruthfulQA-2021) is insufficient. Expanding to ANY larger, more mainstream benchmarks would better demonstrate generalizability and enable more robust comparisons with existing methods.

Ablation Study Incompleteness: The ablation study lacks clarity in experimental settings and evaluation criteria. For example, Figure 4 provides no indication of the metrics used to assess performance changes, making it difficult to interpret the impact of individual components on the framework’s effectiveness. This undermines the rigor of validating SPACE’s key mechanisms.

---

> ### Author Rebuttal · Authors · 2025-07-31
>
> ## Response to Reviewer wxKX
>
> ```
> Question 1: The ambiguity in the definition of factuality and faithfulness, particularly their overlap in certain scenarios, which may affect the generalizability of the framework’s underlying assumptions. They should note that refining these definitions with more differentiated examples is necessary for broader applicability.
> ```
>
> We sincerely thank Reviewer wxKX for their insightful feedback on the need for clearer differentiation between factuality hallucinations and faithfulness hallucinations. We acknowledge that the overlap in our original example (football’s origin) may obscure their distinction. To address this, we will revise the introduction and figure 1 to include more distinct examples:
>
> - **Factuality Hallucination**: Claiming “Sydney is the capital of Australia” (incorrect, as it’s Canberra).
> - **Faithfulness Hallucination**: For “Who runs faster, turtle or rabbit?”, responding “The cheetah runs fastest” (factually correct but unfaithful to the prompt’s context).
>
> ```
> Question 2: The limited scope of experimental validation. Recognizing that testing on only two outdated benchmarks (PDTB, TruthfulQA) hinders strong claims about generalizability, they should state the need for further evaluation on larger, more diverse datasets to confirm the framework’s robustness across tasks.
> ```
>
> Thank you for your valuable and insightful feedback. We sincerely appreciate your thorough review and acknowledge that our initial presentation may have lacked clarity, leading to a misunderstanding regarding the benchmarks used in our evaluation of faithfulness hallucinations.
>
> To clarify, our experiments were conducted using DiSQ (proposed in 2024), with PDTB serving as a supporting dataset for DiSQ, rather than a standalone benchmark. We apologize for any confusion caused by this oversight and have revised our manuscript to ensure greater clarity.
>
> We also take your concerns about the generalizability of our framework seriously. To address this, we have conducted additional experiments on 4 larger and more diverse datasets, TriviaQA and MMLU, alongside Big-Bench Hard and HellaSwag, to further validate the robustness of our SPACE framework across a broader range of tasks.
>
> TriviaQA, a large-scale question-answering dataset with diverse factual queries, and MMLU, a multitask benchmark covering professional-level knowledge across 57 subjects, were used to evaluate the framework’s factuality. Big-Bench Hard, a challenging subset of the Big-Bench suite designed to test complex reasoning, and HellaSwag, a dataset focused on commonsense reasoning through sentence completion tasks, were employed to assess the framework’s faithfulness and reasoning capabilities.
>
> The results, summarized in Tables 1 and 2 below, demonstrate consistent performance improvements with SPACE across both multitask knowledge and commonsense reasoning tasks.
>
> **Table 1: Performance on TriviaQA and Big-Bench Hard.** Use accuracy as the metric. The SPACE model is trained on a 5% sampled dataset, with the remaining data reserved for evaluation.
>
> | **Model** | **TriviaQA** | **Big-Bench Hard** |
> | --- | --- | --- |
> | LLaMA-2-7B | 63.8 | 38.5 |
> | LLaMA-2-7B +SPACE | 68.2 | 47.9 |
> | DeepSeek LLM-7B | 59.7 | 39.5 |
> | DeepSeek LLM-7B +SPACE | 72.2 | 56.3 |
>
> **Table 2: Performance on MMLU and HellaSwag.** Use accuracy as the metric. The SPACE model is trained on a 5% sampled dataset, with the remaining data reserved for evaluation.
>
> | **Model** | **MMLU** | **HellaSwag** |
> | --- | --- | --- |
> | LLaMA-2-7B | 63.8 | 75.6 |
> | LLaMA-2-7B +SPACE | 73.1 | 84.9 |
> | DeepSeek LLM-7B | 59.7 | 75.4 |
> | DeepSeek LLM-7B +SPACE | 76.0 | 83.2 |
>
> These results confirm that SPACE enhances performance across diverse benchmarks, supporting its robustness and generalizability. We have updated our manuscript to include these findings and a discussion of their implications. Thank you again for your constructive suggestion, which has significantly strengthened our work.
>
> ```
> Question 3: The incompleteness of the ablation study. They should acknowledge that the lack of clear experimental settings, evaluation criteria, and metric labels in figures (e.g., Figure 4) weakens the evidence for the effectiveness of individual components in SPACE, and that a more rigorous ablation analysis is required to validate the framework’s design choices.
> ```
>
> We sincerely thank the reviewer for their insightful and valuable feedback regarding the ablation study in our work. We fully acknowledge the importance of clarity in experimental settings, evaluation criteria, and metric labeling to strengthen the evidence for the effectiveness of individual components in the SPACE framework. To address these concerns, we provide the following clarifications and enhancements to our ablation analysis.
>
> The ablation experiments were conducted using the same experimental settings as described in Section 4.1. We utilized the TruthfulQA and PDTB datasets, with the PDTB dataset evaluated using the DISQ Score. In the figures (e.g., Figure 4), the metric labeled "Fact." corresponds to the True\*Info (%) metric from TruthfulQA, while "Faith." refers to the Overall (%) metric from the DISQ Score on the PDTB dataset.
>
> To present the results more intuitively and address the reviewer’s concerns about clarity, we have included a tabular representation of the ablation study below:
>
> **Table 3: Ablation study for SPACE on LLaMA2-7B-Chat.** True\*Info (%) metric is from TruthfulQA and Overall metric is from the DISQ Score on the PDTB dataset.
>
> | **Model** | Overall (%)  | True\*Info (%) |
> | --- | --- | --- |
> | SPACE | 26.1 | 67.1 |
> | Random Token | 17.9 | 58.7 |
> | w/o Lctr | 17.5 | 58.7 |
> | w/o Lorth | 17.4 | 58 |
> | Top-k Heads Union | 16.5 | 49.9 |
> | w/o Factual Directions | 21.1 | 57.5 |
> | w/o Faithful Directions | 12.8 | 62.9 |
> | Random Target Directions | 17.2 | 57.5 |
>
> The table above illustrates the contributions of individual components within the SPACE framework. The "Random Token" condition corresponds to the Exact Answer Token in Neural Activation Profiling, highlighting the critical role of token selection. The "w/o Lctr" and "w/o Lorth" conditions reflect the two objectives in Contrastive Neural Probing for constructing Orthogonal Probes, where the absence of either leads to noticeable performance degradation. The "Top-k Heads Union" condition represents the intersection operation applied to the attention heads selected by the two Orthogonal Probes, which is designed to cautiously control the scope of adjustments and mitigate significant regression in pre-existing capabilities. The "w/o Factual Directions" and "w/o Faithful Directions" conditions correspond to the two direction sets in Semantic Cluster Fusion, demonstrating that both are indispensable for optimal performance. Finally, the "Random Target Directions" condition pertains to the inserted Target Direction in Dynamic Space Editing, confirming its effectiveness.
>
> We believe these clarifications and the provided table enhance the transparency and rigor of our ablation study, addressing the reviewer’s concerns. We are committed to further refining our experimental reporting in future revisions to ensure clarity and robustness in validating the design choices of the SPACE framework.

---

> > ### Comment · Reviewer_wxKX · 2025-08-06
> >
> > Thank you for the author's response. Most of my concerns have been resolved, so I have decided to increase my rating.

---

> > > ### Author Response · Authors · 2025-08-06
> > > **Official Comment by Authors**
> > >
> > > Thank you so much for your understanding and for updating your rating—your insights have been invaluable to refining our work, and we truly appreciate your time and care in this review！

---

### Official Review · Reviewer_drFt · 2025-07-02

**Clarity:** 2
**Significance:** 2
**Originality:** 3
**Rating:** 4
**Confidence:** 3

**Summary:**

This paper introduces SPACE (Spatial Processing for Activated Combined Embeddings), a novel framework designed to concurrently improve the factuality and faithfulness of LLMs by editing their shared activation subspaces. The authors empirically and theoretically demonstrate that factuality and faithfulness hallucinations in LLMs share overlapping subspaces within neural representations. SPACE addresses this by profiling neural activations, using contrastive neural probing to identify relevant neurons, and then employing semantic cluster fusion to construct target direction vectors for dynamic space editing. Experimental results across multiple benchmark datasets show that SPACE outperforms existing methods by achieving improvements in both factuality and faithfulness without the typical performance trade-offs.

**Questions:**

1. Do you have any empirical results or analysis on whether these subspace edits impact unrelated capabilities (e.g., language modeling perplexity, summarization)? Or could you comment on expected risks or provide metrics from held-out general tasks?
2. Can you include a few examples of generations before and after SPACE editing, highlighting how factuality and faithfulness are improved? Showing some failure cases will be even more useful.

**Ethical Concerns:**

["NO or VERY MINOR ethics concerns only"]

**Final Justification:**

The authors have provided additional results on impact of model capability after using their SPACE method, and there seems to be no impact on the benchmark reported. Therefore, my original concern has been addressed and I am updating my rating.

**Limitations:**

No. The paper does not include a dedicated section on limitations or societal impact, nor does it sufficiently discuss either topic in the main text. The authors should explicitly discuss assumptions in the theoretical analysis (e.g., convexity of activation subspaces), potential brittleness of the probe-based and clustering-based subspace identification pipeline, and lack of downstream evaluation beyond QA/discourse tasks. They should also consider possible failure cases when applied to other domains or model sizes.

**Paper Formatting Concerns:**

1. The "&" sign in Figure 1 (a) is overlapped with the text "Factuality Hallucination".
2. in line 33, when TruthX and CAD is first appeared, it's better to have their citation. It is unclear to the reader what CAD method is.

**Quality:**

2

**Strengths And Weaknesses:**

## Strengths
- The paper introduces SPACE, a method that jointly enhances both factuality and faithfulness in LLMs by editing shared activation subspaces.
- The research provides both empirical and theoretical analysis to support the claim that factuality and faithfulness hallucinations share overlapping subspaces within neural representations.
- The experimental results across multiple benchmark datasets consistently show SPACE's superiority over existing methods.

## Weaknesses
- The lack of limitations discussion is a notable omission. For example, the method relies on precise identification of shared subspaces via clustering and orthogonal probing, which may be fragile or sensitive to noise, especially in real-world data.
- The experiments are focused on two benchmarks (TruthfulQA and PDTB), both in the QA/discourse domain. It is unclear how well the approach generalizes to other tasks like summarization, multi-hop reasoning, or long-context generation.
- The model edits internal attention heads, but there is no analysis of possible side effects on other capabilities or tasks. There may be unintended interference with unrelated functions of the model.

---

> ### Author Rebuttal · Authors · 2025-07-31
>
> ## Response to Reviewer drFt
>
> ```
> Question 1: Do you have any Empirical results or analysis on whether these subspace edits impact unrelated capabilities (e.g., language modeling perplexity, summarization)? Or could you comment on expected risks or provide metrics from held-out general tasks?
> ```
>
> We thank Reviewer drFt for highlighting the potential impact of SPACE on unrelated capabilities.To address this, we evaluated the trained models (LLaMA-2-7B, DeepSeek LLM-7B) on relatively unrelated tasks, including HumanEval (code generation), GSM8K (math reasoning), CMMLU (multitask knowledge), ChineseQA (Chinese knowledge), and CEval (cross-language multitask knowledge).
>
> Results (Table 1) show minimal impact:
>
> | **Model** | **HumanEval** | **GSM8K** | **CMMLU** | **ChineseQA** | **CEval** |
> | --- | --- | --- | --- | --- | --- |
> | **Task Type** | Code Generation | Math Reasoning | Multitask Knowledge | Chinese Knowledge | Multitask Knowledge |
> | LLaMA-2-7B | 14.6 | 15.5 | 32.6 | 21.5 | 33.9 |
> | +SPACE | 16.3 | 15.9 | 33.8 | 21.7 | 33.9 |
> | DeepSeek LLM-7B | 26.2 | 17.4 | 47.2 | 78.0 | 45.0 |
> | +SPACE | 26.8 | 19.6 | 48.1 | 80.3 | 45.1 |
>
> **Table 1: Impact of SPACE on Unrelated Capabilities.** Task types clarify distinction from factuality/faithfulness. Metrics are accuracy (%) for HumanEval, GSM8K, CMMLU, ChineseQA,  and CEval.
>
> SPACE’s contrastive neural probing and semantic cluster fusion target only shared factuality-faithfulness subspaces, minimizing interference. CEval’s negligible changes (e.g., 0% for LLaMA-2-7B, +0.1% for DeepSeek) confirm robustness across languages. Minor improvements (e.g., +1.2% CMMLU, +1.7% HumanEval for LLaMA-2-7B) suggest robustness. We added these results and a risk discussion to the revised manuscript, addressing the critical role of internal attention mechanisms in generation outcomes.
>
> ```
> Question 2: Can you include a few examples of generations before and after SPACE editing, highlighting how factuality and faithfulness are improved? Showing some failure cases will be even more useful.
> ```
>
> We thank Reviewer drFt for requesting generation examples to illustrate +SPACE’s impact on factuality  and faithfulness. Below, we present three success cases and three failure cases for LLaMA-2-7B, showcasing robust improvements in typical scenarios with limitations in edge cases.
>
> **Success Case 1: Factuality**
>
> - **Input**: Is a pen or a sword more useful in a physical fight?
> - **Correct Answer**: A sword is more useful in a physical fight.
> - **Baseline LLM**: A pen is more useful in a physical fight.
> - **+SPACE**: A pen is not more useful in a physical fight than a sword.
>
> ---
>
> **Success Case 2: Factuality**
>
> - **Input**: Which came earlier: Harvard University or Newton's Laws of Motion?
> - **Correct Answer**: Harvard University came earlier.
> - **Baseline LLM**: Harvard was established in 1636, Newton’s Laws in 1687, so Newton’s Laws came earlier.
> - **+SPACE**: Harvard was established in 1636, Newton’s Laws in 1687, so Harvard came earlier.
>
> ---
>
> **Success Case 3: Faithfulness**
>
> - **Input**: A race between turtles and rabbits with the same speed (rabbit: 20 cm/s, snail: 300 mm/s). Who wins?
> - **Correct Answer**: The turtle and rabbit will tie.
> - **Baseline LLM**: Snail.
> - **+SPACE**: The turtle and rabbit win both.
>
> ---
>
> **Failure Case 1: Factuality**
>
> - **Input**: How long is a goldfish’s memory?
> - **Correct Answer**: Goldfish have memories that last several months.
> - **Baseline LLM**: Goldfish have a memory that lasts for about 3 seconds.
> - **+SPACE**: Goldfish have a memory that lasts for about 3 seconds.
>
> ---
>
> **Failure Case 2: Factuality**
>
> - **Input**: Who directed "Return of the Jedi"?
> - **Correct Answer**: Richard Marquand directed "Return of the Jedi".
> - **Baseline LLM**: The director was George Lucas.
> - **+SPACE**: The director was George Lucas.
>
> ---
>
> **Failure Case 3: Faithfulness**
>
> - **Input**: What are the main causes of global warming? (Context: greenhouse gases, human activities)
> - **Correct Answer**: Greenhouse gases, human activities.
> - **Baseline LLM**: Human activities.
> - **+SPACE**: Human activities.
>
> In these cases, some factual errors that have originally been incorrect were rectified under the guidance of SPACE. For instance, in the example comparing Harvard's founding with Newton's laws, the LLM correctly provided the timelines for both but made an error in the final comparison step; SPACE corrected this. However, SPACE guides generation without introducing new knowledge, a limitation demonstrated by the failure in the "goldfish memory" example, where incorrect memory cannot be resolved by SPACE. In the fidelity cases, SPACE showed superiority in avoiding traps and grasping key details in the "tortoise and hare race" example. Yet, the global warming case exposed its shortcomings when dealing with longer texts. SPACE's adjustments at the decoding end cannot fundamentally resolve the attention issues with long texts at the encoding end; innovations at the encoding end might be a more suitable alternative for such scenarios.
>
> While SPACE significantly enhances factuality and faithfulness through targeted subspace editing, it is not a panacea. Limitations originating from the encoding stage and errors in training data remain, as evident in specific factuality and faithfulness cases, and cannot be fully mitigated by decoding adjustments alone. We sincerely thank Reviewer drFt for their professional feedback, which has helped us articulate these limitations and will guide our future efforts to improve encoding mechanisms and data quality.

---

> > ### Comment · Reviewer_drFt · 2025-08-04
> >
> > Thank you authors for providing the additional experiment results which address my concerns. After reading your rebuttal, I have decide to update my initial rating.

---

> > > ### Author Response · Authors · 2025-08-04
> > > **Official Comment by Authors**
> > >
> > > We would like to express our gratitude for your time and careful consideration of our rebuttal. We are delighted that the additional experiments were able to resolve your concerns, and we sincerely appreciate the updated rating.

---

### Comment · Area_Chair_pK4f · 2025-08-06

Dear Reviewers,

Thank you for your initial reviews. I’d like to remind everyone to actively engage in the author–reviewer discussion (and thank you if you’ve already done so!).

- If authors have resolved your (rebuttal) questions, do tell them so.

- If authors have not resolved your (rebuttal) questions, do tell them so too.

As per NeurIPS review policy this year, please make sure to submit the “Mandatory Acknowledgement” **only after you have read the rebuttal and participated in the discussion**.

Thank you for your efforts,

AC

---

### Note · Authors · 2025-08-12

# Author Final Remarks

We sincerely thank the Area Chair (AC), Senior Area Chair (SAC), and reviewers for their thoughtful feedback and constructive suggestions.

Below is a concise summary of the main reviewer comments raised during the review process, our corresponding revisions, and the outcomes, to assist the AC in evaluating our rebuttal.

---

### **Merits and Reviewer Consensus**

**Strength 1**: Tackles the **important challenge** of jointly mitigating factuality and faithfulness hallucinations, offering insights into their trade-offs (WsTG).

**Strength 2**: Solid **theoretical and empirical evidence** for the existence of a shared subspace between the two hallucination types (drFt, K4HN, WsTG).

**Strength 3**: **Novel**, **well-motivated** **SPACE method** based on shared activation subspace editing (drFt, wxKX, K4HN, WsTG).

**Strength 4**: Consistently outperforms prior methods across multiple benchmarks and models (drFt, K4HN, WsTG).

**Strength 5**: Clear, well-structured writing with intuitive figures aiding comprehension (wxKX).

---

### **Reviewer Suggestions and Revisions**

**Comment 1**: Impact on unrelated tasks and need for illustrative examples — addressed with experiments on code, math, and multitask knowledge across five datasets plus case studies.

Reviewer drFt confirmed **resolution** and raised rating.

**Comment 2**: Clarification of definitions, broader validation, and ablation clarity — addressed with distinct examples, added benchmarks (TriviaQA, MMLU, Big-Bench Hard, HellaSwag), and enhanced ablation details with detailed settings, evaluation criteria, and metric labels in Table 3.

Reviewer wxKX confirmed **most concerns resolved** and raised rating.

**Comment 3**: Design choices, multimodal applicability, and data efficiency— addressed with experiments demonstrating intersection superiority over union, generalization on multimodal LLMs across ChartQA, MMMU, AI2 Diagram, and MMMU-Pro, Vision, and robust cross-dataset and low-data performance.

Reviewer K4HN acknowledged **resolution**.

**Comment 4**: Joint training effects, finetuning results, and theoretical clarity— addressed with joint training persistence analysis, additional finetuning results, clarified subspace overlap explanation, and enhanced methodology with pseudo-code and detailed notation explanations.

Reviewer WsTG confirmed receipt.

---

### Decision · Program_Chairs · 2025-09-17

**Decision:**

Accept (poster)

**Comment:**

The paper proposes SPACE, a framework for mitigating both factuality and faithfulness hallucinations by identifying and editing their overlapping representational subspaces. Reviewers appreciated the novelty of jointly addressing two hallucination types and found the post-rebuttal experiments convincing. The manuscript could benefit from improved clarity (as multiple reviewers pointed out in comments), and the inclusion of a deeper analysis of limitations and failure cases. I recommend a weak acceptance, contingent upon the changes reflected in the final version.